# Effect of a Second Pregnancy on the HPV Serology in Mothers Followed Up in the Finnish Family HPV Study

**DOI:** 10.3390/v15102109

**Published:** 2023-10-18

**Authors:** Helmi Suominen, Nelli Suominen, Kari Syrjänen, Tim Waterboer, Seija Grénman, Stina Syrjänen, Karolina Louvanto

**Affiliations:** 1Department of Obstetrics and Gynecology, Faculty of Medicine and Health Technology, Tampere University, 33520 Tampere, Finland; karolina.louvanto@tuni.fi; 2Department of Obstetrics and Gynecology, Turku University Hospital and University of Turku, 20014 Turku, Finland; nelli.t.suominen@utu.fi (N.S.); seija.grenman@tyks.fi (S.G.); 3Department of Obstetrics and Gynecology, Vaasa Central Hospital, 65130 Vaasa, Finland; 4SMW Consultants, Ltd., 21620 Kaarina, Finland; kasyrja@saunalahti.fi; 5Division of Infections and Cancer Epidemiology, German Cancer Research Center (DKFZ), 69120 Heidelberg, Germany; t.waterboer@dkfz-heidelberg.de; 6Department of Pathology, Turku University Hospital, 20014 Turku, Finland; stisyr@utu.fi; 7Department of Oral Pathology and Oral Radiology, Institute of Dentistry, Faculty of Medicine, University of Turku, 20014 Turku, Finland; 8Department of Obstetrics and Gynecology, Tampere University Hospital, 33100 Tampere, Finland

**Keywords:** human papillomavirus, HPV, serology, mother, child

## Abstract

The impact of pregnancy on human papillomavirus (HPV) natural antibody levels is not fully understood. We tested the seroprevalence and levels of HPV 6, 11, 16, 18 and 45 antibodies at different time points among 89 women with a second pregnancy and 238 nonpregnant women during their 36-month followup. All participants were unvaccinated for HPV and pregnant at the enrollment of the study. Serum samples were collected from the mothers at baseline and at the 12-month, 24-month, and 36-month followup visits. No statistically significant differences in mean antibody levels were observed in women who developed a second pregnancy compared to their nonpregnant counterparts. Between these two groups, statistically significant differences in serostatus were observed, particularly if the second pregnancy was ongoing at the 24-month timepoint. Accordingly, women with a second pregnancy were more likely to be seronegative for HPV 6, 11, 18, and 45 as compared to the nonpregnant women, the reverse being true for HPV16. In contrast, the women with an ongoing second pregnancy showed a higher prevalence of HPV16 seropositivity at the 36-month followup. These data suggest that a second pregnancy does not seem to have a major impact on the levels of HPV antibodies, but it might influence the serological outcomes.

## 1. Introduction

Acquisition of human papillomavirus (HPV) infection is common especially in young adults [1]. It has been estimated that nearly all individuals become infected with some HPV type at least once in a lifetime [2]. For most individuals, mucosal HPV infections are transient and never result in viral persistence or clinical progression. As HPV infections are most prevalent among young adults, the possible impact of pregnancy on the dynamics of HPV infection is of substantial interest [3].

There is some evidence that pregnancy-related hormonal changes and immunosuppression could have an impact on the prevalence of HPV infection and viral persistence [4,5,6,7]. So far, the effect of pregnancy on HPV antibodies is more unclear, as studies have reported contradictory results. In addition, the antibody response resulting from a naturally acquired HPV infection is known to vary remarkably. Confirmed seropositivity can be regarded as a sign of previous exposure to HPV, although seroconversion does not occur in all HPV-infected individuals [8]. Optimally, antibodies might offer some protection against subsequent HPV infections. In a study evaluating genital HPV infection, it was suggested that HPV antibodies acquired as a result from a natural HPV infection offer only modest protection against a genital HPV reinfection [9]. As for HPV antibodies in pregnancy, it has not been firmly established whether pregnancy decreases or increases HPV antibody levels and HPV seropositivity in general.

Followup data on previously pregnant women are lacking to assess whether becoming pregnant for the second time influences HPV seropositivity and HPV antibody levels. In our previous report from this same cohort, the second pregnancy had little independent impact on the carriage and persistence of oral and cervical high-risk HPV infections [3], but the data on HPV serology were not available at that time. In the present study, we analyzed the seroprevalence for HPV types 6, 11, 16, 18, and 45 in women who developed a second pregnancy during followup in the prospective Finnish Family HPV Study cohort.

## 2. Materials and Methods

### 2.1. Participants

The Finnish Family HPV (FFHPV) Study is a longitudinal cohort study, designed to analyze the dynamics of HPV transmission within regular families (mother, father, and the index offspring), conducted since 1998 jointly by the Department of Obstetrics and Gynecology (Turku University Hospital) and the Institute of Dentistry (Faculty of Medicine, University of Turku). The original study protocol and its amendments were approved by the Research Ethics Committee of Turku University Hospital (#3/1998, #2/2006, 45/180/2010). Originally, 329 pregnant women in their third trimester and their 331 newborns (including two twins) were enrolled in the study between 1998 and 2001, and written informed consent was obtained from each participant. The study design as well as the details of the participants have been given in a series of previous reports [10,11,12]. None of the participants had received a prophylactic HPV vaccination, and therefore all the antibodies resulted from a naturally acquired HPV infection. The present analysis includes, as the study group, all 89 women who became pregnant for the second time during the followup and, as the reference group, the remaining 238 women who did not have a second pregnancy during the followup. Two women out of the original 329 women were lost to followup; so, they were excluded from the present study.

### 2.2. Serology

Serum samples were collected from the mothers at the baseline (36 weeks of pregnancy) of the study and at the 12-month, 24-month, and 36-month followup visits. Antibodies to the major capsid protein L1 of HPV6, 11, 16, 18, and 45 were analyzed with multiplex HPV serology based on glutathione S-transferase fusion-protein capture on fluorescent beads as previously described [13]. For all HPV types, seropositivity was defined as the median fluorescence intensity (MFI) > 200 or >400 (stringent cutoff).

### 2.3. Statistical Analyses

Frequency tables were analyzed using the χ^2^ test or the Fisher’s exact test for categorical variables (seropositivity +/−). Differences in the means of continuous variables (MFI titers) were analyzed using ANOVA (analysis of variance) after controlling for their normal distribution. The Kolmogorov–Smirnov test was used for normality. The two groups were also compared by the distribution of the potential HPV-associated covariates recorded by a detailed questionnaire at the study enrollment. SPSS 26.0.1 (IBM, Armonk, NY, USA) and STATA/SE 16.1 (Stata Corp., College Station, TX, USA) software packages were used. All statistical tests performed were two-sided and declared significant at the *p*-value level of <0.05.

## 3. Results

The mean antibody levels of the HPV6, HPV11, HPV16, HPV18, and HPV45 seropositive women at baseline and during the followup visits were stratified by their status of second pregnancy and are shown in Figure 1. Seropositive women were categorized by either having an ongoing second pregnancy at the time of the followup visit, having a second pregnancy at some other timepoint, or not having a second pregnancy at all during the followup. These two separate groups in the figure for the women with second pregnancy were used to illustrate the effect of the ongoing pregnancy at the timepoint compared to another time point in the longitudinal followup. As for HPV18 antibodies at the baseline of the study, the mean levels (MFI mean ± SD) were 125 (±117) for women who developed a second pregnancy, and 199 (±217) for women who did not have a second pregnancy during the followup (*p* = 0.021), but no other significant differences were observed later on between these groups. The mean antibody levels to HPV6 and HPV16 showed some variation over time between these groups, but no statistically significant differences were observed. As for antibodies to HPV16, the mean MFI at 12 months was higher in those women who had an ongoing pregnancy at that 12-month timepoint with a mean MFI 1164 (±2003), while the corresponding mean MFI values were 803 (±971) for those who had a second pregnancy at some other timepoint than at 12 months and 611 (±1217) for those who did not develop a second pregnancy during the followup (*p* = 0.209).

All 327 women included in this study were further stratified by their HPV6, HPV11, HPV16, HPV18, and HPV45 serostatus (seropositive or seronegative), as well as their serostatus during the followup as related to the timing of the second pregnancy (12 months, 24 months, and 36 months) (Table 1). Most of the significant differences in serostatus were observed among the cases when the second pregnancy was ongoing at the 24-month timepoint. This difference between the 24-month pregnant and nonpregnant women applies to their baseline seropositivity for (1) HPV6 (*p* = 0.008) and (2) HPV11 (*p* = 0.002). As to the HPV11 serostatus between the 24-month pregnant and nonpregnant women, significant differences were observed in all timepoints except in the 36-month followup visit. In addition, the HPV6 and HPV18 serostatus at 36 months was different between the 24-month pregnant and nonpregnant women; *p* = 0.042 and *p* = 0.005, respectively. When the second pregnancy was ongoing at the 36-month followup visit, statistically significant differences between pregnant and nonpregnant women were observed only in the 24-month followup visit serostatus: for HPV6 and HPV16, *p* = 0.021 and *p* = 0.04, respectively. No such differences were observed when the second pregnancy was ongoing at the 12-month followup visit.

The serostatus of HPV6, HPV11, HPV16, HPV18, and HPV45 at the baseline and during the 12-, 24- and 36-month followup visits, with both MFI > 200 and stringent MFI > 400 seropositivity cutoffs, are shown in Appendix A. At the baseline of the study, only 44.9% (n = 40) of the women who developed a second pregnancy were HPV6 seropositive (MFI > 200) as compared with 58.4% (n = 139) of those who did not develop a second pregnancy (*p* = 0.034). Similarly, 6.7% (n = 6) of the women with their second pregnancy were HPV11 seropositive (with stringent MFI > 400) at baseline, as compared with 15.1% (n = 36) of those who did not develop a second pregnancy (*p* = 0.043). The two groups also differed significantly in their baseline seropositivity (MFI > 200) for HPV18 (*p* = 0.013). In addition, significant differences between women with and women without a second pregnancy were observed in their HPV18 (MFI > 200) serostatus at the 12-month and 36-month followup visits: *p* = 0.002 and *p* = 0.044, respectively.

The demographic and clinical data of the women are shown in Table 2. Women with a second pregnancy and nonpregnant women differed significantly in their marital status, number of deliveries, number of lifetime sexual partners, history of STDs, and previous contraception method used. A smaller number of women with a second pregnancy were single when compared to nonpregnant women: 2.3% vs. 8.9% (*p* = 0.045). Women with a second pregnancy had fewer previous deliveries than the nonpregnant women: 11.6% vs. 30.9% for two or more deliveries (*p* = 0.002). This also explains why more condoms and oral contraceptives were used in the past among women with a second pregnancy compared to the nonpregnant women where no contraception was the most common used method of contraception (*p* = 0.032). Also, women with a second pregnancy had fewer lifetime sexual partners (*p* = 0.038) but having a positive history of STDs was slightly more common in this group (*p* = 0.038) as compared to their nonpregnant counterparts. The mean age with SD was 25.4 ± 3.4 for the women with a second pregnancy and 25.5 ± 3.4 for those who did not develop a second pregnancy (*p* = 0.718).

## 4. Discussion

To our knowledge, this is the first study evaluating the effect of a second consecutive pregnancy on naturally acquired HPV antibodies in a longitudinal setting. Our results indicate only slight differences in the mean antibody levels to HPV6, HPV11, HPV16, HPV18, and HPV45 between women who developed a second pregnancy and those women who did not develop a second pregnancy during the followup. Our earlier study on HPV serology in this cohort not stratified by the second pregnancy showed that HPV seroprevalence was lowest at the entry of the study when all women were pregnant at their third trimester. Both low-risk and high-risk HPV seropositivity were significantly associated with the age at onset of sexual activity, the number of sexual partners until 20 years of age, the lifetime number of sexual partners, and the history of genital warts [12].

In the present series, the mean antibody levels to HPV6, HPV11, HPV18, and HPV45 appeared somewhat lower in those women who developed a second pregnancy, but in contradiction, this was not the case for HPV16, in which higher antibody levels were recorded as well as with individual mean values for HPV6 at the baseline and for HPV45 at the 12-month followup visit. As for the differences in HPV6 antibody levels between the two groups of women in the baseline, our results indicate that there could be some baseline differences that affect HPV6 antibody levels between these two groups, and it is important to notice that this difference is not related to the second pregnancy that takes place later on. In contrast, the general trend in the mean antibody levels of our data was a slight decrease when comparing the women with second pregnancy to those without, and this trend was also seen with HPV45 in all timepoints investigated except for one. One possible explanation for this is that antibody response provoked by a naturally acquired HPV infection is known to vary between different individuals; therefore, this could affect our results. In addition, there is some controversy on the stability of HPV antibodies, although HPV IgG antibodies are believed to be relatively stable over time [11].

In this study, we observed that women who developed a second pregnancy during the followup differed from those who did not develop a second pregnancy in terms of their marital status, the number of deliveries, the number of lifetime sexual partners, the history of STDs, and the contraception method used previously (Table 2). These variables are considered as risk factors for HPV infection [5,14,15] and therefore represent potential confounding factors. Women with no second pregnancy reported more lifetime sexual partners and deliveries and no contraception as compared to women who had their second pregnancy during the followup. These background differences might predict lower HPV antibody levels and a higher proportion of seronegative outcomes among the women with second pregnancy, due to less exposure to HPV-related cofactors.

In general, the total IgG level has been suggested to be lower during pregnancy [16]. The activation of B-lymphocytes has been shown to continue from becoming pregnant to the postpartum period, affecting the antibody secretion of different immunoglobulin classes [17]. Studies on different IgG subclasses have yielded contradictory results; some studies suggest that IgG subclass IgG1 is stable during pregnancy, whereas some suggest its levels are higher during pregnancy, and IgG3 levels are measured to be higher, but IgG2 and IgG4 levels have been thought to remain stable [16,17]. Hemodilution is thought to be one of the causes of lower total IgG levels during pregnancy; however, the suppression of cell-mediated immunity, the loss of protein in urine, the placental transfer of IgGs to the fetus, and hormonal changes, might contribute to this [16,18]. Our earlier study on this same cohort implied that the IgG antibody levels were lower at the baseline, and an increase in the antibody levels was seen after pregnancy [11]. Accordingly, one study assessing serological responses to HPV16 E4, E6, and E7 proteins in pregnant women suggested that the humoral immune response against HPV infections is reduced during pregnancy [19]. In our previous analysis from the FFHPV cohort, IgG antibodies to HPV16 L1 in serum were lower during pregnancy, but the serum IgA antibodies showed a different pattern [20]. However, the possible effect of differing HPV prevalence between different HPV serology studies assessing HPV antibody levels in pregnant women must be taken into consideration. According to one meta-analysis, the overall HPV prevalence in pregnant women varies by study region, age, and HPV type, and its results demonstrated that pregnant women are more susceptible to HPV infection than their nonpregnant counterparts [7].

Our data suggest that a second pregnancy does not increase HPV seropositivity, and the observed changes in mean antibody levels and differences in serostatuses could result from differences in the women’s background or immunological factors not the second pregnancy at the followup visit itself. As for other DNA viruses and their significance in pregnancy, nearly all human herpesviruses (HHVs) have been shown to infect cells at the fetal–maternal interface without crossing the placental barriers [21]. In a study investigating IgG antibody titers to Epstein–Barr virus infection in pregnant women, the overall antibody levels declined during late pregnancy, and latent viral reactivation was observed to occur due to the potential stress-induced immune dysregulative state especially in racial disparities [22]. With herpes simplex viruses, the overall seroprevalence for both HSV-1 and HSV-2 is relatively high in pregnant women, and the presence of HSV IgG antibodies in relation to the timing of viral reactivation is associated with pregnancy and neonatal complications [23]. Lastly, with the cytomegalovirus infections, pre-existing maternal antibodies to CMV may act as a protective factor against congenital CMV infection, but viral reactivation or new maternal infection with another virus strain may lead to fetal infection [24].

As many previous studies on HPV serology have had a cross-sectional design, one of the strengths of this study was the use of a longitudinal design that allowed the assessment of the effects of a second consecutive pregnancy on HPV serology with five different HPV genotypes. This study has a unique design on the subject that has not been previously addressed from the viewpoint of a second pregnancy’s effect on HPV serology. The women’s HPV serostatus and antibody levels were measured at four different timepoints (baseline, 12, 24, and 36 months), but the possible impact of the actual gestational length of the second pregnancy at each time point was not taken in consideration, which is one factor that could also affect our results. Moreover, a known limitation in all serological studies is that not all individuals seroconvert [8], even in the case of a persistent HPV infection, and this limitation must be taken into consideration when interpreting our results. In addition, another limitation in serology studies is that currently there is no golden standard method for assessing HPV antibodies, although efforts have begun in order to standardize HPV serology assays [25]. In this particular study, we used the multiplex serology assay, which is useful in evaluating cumulative HPV infection, although it is not a reliable marker of immune protection, as it does not differentiate between neutralizing and non-neutralizing antibodies [26]. Furthermore, the significance of these measured antibodies in protecting the women of our cohort against future HPV infection is uncertain.

To conclude, HPV antibody levels and HPV serostatus showed only slight variations during the second pregnancy according to our data in the FFHPV cohort. Women with a second pregnancy were less likely to be seropositive for HPV6, HPV11, HPV16, HPV18, and HPV45 as compared to women without a second pregnancy. However, apart from pregnancy, more attention needs to be paid on cofactors that might also impact the serological outcomes.

## Figures and Tables

**Figure 1 viruses-15-02109-f001:**
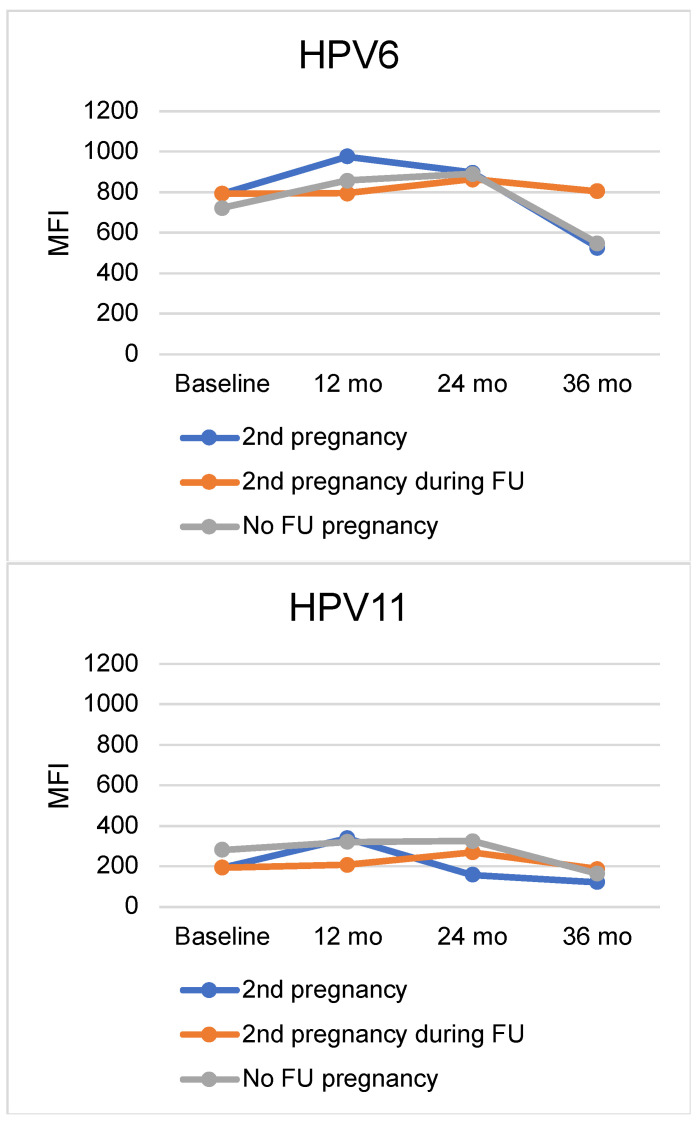
Mean MFI levels of HPV6, HPV11, HPV16, HPV18, and HPV45 antibodies at baseline and at the 12-month, 24-month, and 36-month followup visits among seropositive women stratified by the status of their second pregnancy. The blue line depicts the mean MFI values of those who had a second pregnancy at each specific followup timepoint, the orange line depicts those who had a second pregnancy at some other timepoint during the followup time, and the gray line depicts those who did not have a second pregnancy at all. Abbreviations: MFI = median fluorescence intensity, mo = months, FU = follow-up.

**Table 1 viruses-15-02109-t001:** HPV6, HPV11, HPV16, HPV18, and HPV45 serostatus among the 327 women from the Finnish Family HPV study during the 36 months of followup stratified by the timing of the second pregnancy (12 mo, 24 mo, and 36 mo). Significant differences in the seroprevalence of women with a second pregnancy at the given time point and the rest of the cohort are bolded. When the second pregnancy was ongoing at 12 months, no significant differences were observed, but significant differences were observed with a second pregnancy at the 24- or 36-month timepoint.

		Baseline		12 Months		24 Months		36 Months	
		Yes	No	Yes	No	Yes	No	Yes	No
		n (%)		n (%)		n (%)		n (%)	
2nd Pregnancy at 12 mo (n = 27)									
HPV6	Seropositive	17 (63.0)	162 (54.0)	18 (66.7)	178 (69.3)	19 (73.1)	159 (67.4)	14 (51.9)	126 (53.8)
	Seronegative	10 (37.0)	138 (46.0)	9 (33.3)	79 (30.7)	7 (26.9)	77 (32.6)	13 (48.1)	108 (46.2)
HPV11	Seropositive	8 (29.6)	62 (20.7)	9 (33.3)	69 (26.8)	7 (26.9)	55 (23.3)	6 (22.2)	33 (14.1)
	Seronegative	19 (70.4)	238 (79.3)	18 (66.7)	188 (73.2)	19 (73.1)	181 (76.7)	21 (77.8)	201 (85.9)
HPV16	Seropositive	10 (37.0)	99 (33.0)	12 (44.4)	104 (40.5)	10 (38.5)	78 (33.1)	8 (29.6)	66 (28.2)
	Seronegative	17 (63.0)	201 (67.0)	15 (55.6)	153 (59.5)	16 (61.5)	158 (66.9)	19 (70.4)	168 (71.8)
HPV18	Seropositive	5 (18.5)	61 (20.3)	5 (18.5)	70 (27.2)	7 (26.9)	52 (22.0)	8 (29.6)	53 (22.6)
	Seronegative	22 (81.5)	239 (79.7)	22 (81.5)	187 (72.8)	19 (73.1)	184 (78.0)	19 (70.4)	181 (77.4)
HPV45	Seropositive	3 (11.1)	28 (9.3)	4 (14.8)	28 (10.9)	5 (19.2)	19 (8.1)	2 (7.4)	18 (7.7)
	Seronegative	24 (88.9)	272 (90.7)	23 (85.2)	229 (89.1)	21 (80.8)	217 (91.9)	25 (92.6)	216 (92.3)
2nd pregnancy at 24 mo (n = 43)									
HPV6	Seropositive	**15 (34.9) ^a^**	**164 (57.7) ^a^**	26 (60.5)	170 (70.5)	23 (54.8)	155 (70.5)	**16 (38.1) ^b^**	**124 (56.6) ^b^**
	Seronegative	**28 (65.1) ^a^**	**120 (42.3)^a^**	17 (39.5)	71 (29.5)	19 (45.2)	65 (29.5)	**26 (61.9) ^b^**	**95 (43.4) ^b^**
HPV11	Seropositive	**2 (4.7) ^c^**	**68 (23.9) ^c^**	**6 (14.0) ^d^**	**72 (29.9) ^d^**	**3 (7.1) ^e^**	**59 (26.8) ^e^**	3 (7.1)	36 (16.4)
	Seronegative	**41 (95.3) ^c^**	**216 (76.1)^c^**	**37 (86.0) ^d^**	**169 (70.1) ^d^**	**39 (92.9) ^e^**	**161 (73.2) ^e^**	39 (92.9)	183 (83.6)
HPV16	Seropositive	10 (23.3)	99 (34.9)	14 (32.6)	102 (42.3)	10 (23.8)	78 (35.5)	9 (21.4)	65 (29.7)
	Seronegative	33 (76.7)	185 (65.1)	29 (67.4)	139 (57.7)	32 (76.2)	142 (64.5)	33 (78.6)	154 (70.3)
HPV18	Seropositive	4 (9.3)	62 (21.8)	**4 (9.3) ^f^**	**71 (29.5) ^f^**	6 (14.3)	53 (24.1)	**3 (7.1) ^g^**	**58 (26.5) ^g^**
	Seronegative	39 (90.7)	222 (78.2)	**39 (90.7) ^f^**	**170 (70.5) ^f^**	36 (85.7)	167 (75.9)	**39 (92.9) ^g^**	**161 (73.5) ^g^**
HPV45	Seropositive	1 (2.3)	30 (10.6)	**1 (2.3) ^h^**	**31 (12.9) ^h^**	1 (2.4)	23 (10.5)	2 (4.8)	18 (8.2)
	Seronegative	42 (97.7)	254 (89.4)	**42 (97.7) ^h^**	**210 (87.1) ^h^**	41 (97.6)	197 (89.5)	40 (95.2)	201 (91.8)
2nd pregnancy at 36 mo (n = 19)									
HPV6	Seropositive	8 (42.1)	171 (55.5)	12 (63.2)	184 (69.4)	13 (68.4)	165 (67.9)	14 (70.0)	126 (52.3)
	Seronegative	11 (57.9)	137 (44.5)	7 (36.8)	81 (30.6)	6 (31.6)	78 (32.1)	6 (30.0)	115 (47.7)
HPV11	Seropositive	3 (15.8)	67 (21.8)	6 (31.6)	72 (27.2)	**9 (47.4) ^i^**	**53 (21.8) ^i^**	3 (15.0)	36 (14.9)
	Seronegative	16 (84.2)	241 (78.2)	13 (68.4)	193 (72.8)	**10 (52.6) ^i^**	**190 (78.2) ^i^**	17 (85.0)	205 (85.1)
HPV16	Seropositive	8 (42.1)	101 (32.8)	11 (57.9)	105 (39.6)	**11 (57.9) ^j^**	**77 (31.7) ^j^**	8 (40.0)	66 (27.4)
	Seronegative	11 (57.9)	207 (67.2)	8 (42.1)	160 (60.4)	**8 (42.1) ^j^**	**166 (68.3) ^j^**	12 (60.0)	175 (72.6)
HPV18	Seropositive	1 (5.3)	65 (21.1)	4 (21.1)	71 (26.8)	3 (15.8)	56 (23.0)	3 (15.0)	58 (24.1)
	Seronegative	18 (94.7)	243 (78.9)	15 (78.9)	194 (73.2)	16 (84.2)	187 (77.0)	17 (85.0)	183 (75.9)
HPV45	Seropositive	1 (5.3)	30 (9.7)	3 (15.8)	29 (10.9)	3 (15.8)	21 (8.6)	2 (10.0)	18 (7.5)
	Seronegative	18 (94.7)	278 (90.3)	16 (84.2)	236 (89.1)	16 (84.2)	222 (91.4)	18 (90.0)	223 (92.5)

*p*-values = ^a^ 0.008, ^b^ 0.042, ^c^ 0.002, ^d^ 0.028, ^e^ 0.004, ^f^ 0.005, ^g^ 0.005, ^h^ 0.039, ^i^ 0.021, ^j^ 0.04. Abbreviations: MFI = median fluorescence intensity, mo = months. Cutoff value for seropositivity was MFI > 200.

**Table 2 viruses-15-02109-t002:** Demographic and clinical data of the women with a second pregnancy compared to the women that did not have an additional pregnancy during the 36-month followup of the Finnish Family HPV Study cohort. Significant comparisons are bolded.

Variable	2nd Pregnancy	No 2nd Pregnancy	Significance
	n (%)		
Marital status			*p* = 0.045 *
Single	**2 (2.3)**	**18 (8.9)**	
Other (unmarried couple, married, divorced)	**84 (97.7)**	**184 (91.1)**	
Number of deliveries			*p* = 0.002 *
0	**1 (1.2)**	**1 (0.5)**	
1	**75 (87.2)**	**138 (68.7)**	
2	**8 (9.3)**	**54 (26.9)**	
3	**2 (2.3)**	**5 (2.5)**	
4	**0 (0.0)**	**3 (1.5)**	
Age at first intercourse			*p* = 0.179 *
≤13	3 (3.5)	4 (2.0)	
14–16	44 (51.2)	117 (57.9)	
17–19	32 (37.2)	75 (37.1)	
≥20	7 (8.1)	6 (3.0)	
Number of lifetime sexual partners			*p* = 0.038
1–2	**28 (32.9)**	**43 (21.3)**	
3–5	**27 (31.8)**	**64 (31.7)**	
6–10	**20 (23.5)**	**45 (22.3)**	
>10	**10 (11.8)**	**50 (24.8)**	
Number of sexual partners by the age of 20			*p* = 0.262 *
0–2	44 (51.2)	80 (39.6)	
3–5	24 (27.9)	74 (36.6)	
6–10	14 (16.3)	32 (15.8)	
>10	4 (4.7)	16 (7.9)	
Frequency of intercourse, n/month			*p* = 0.103 *
0–1	0 (0.0)	7 (3.5)	
2–4	27 (31.4)	59 (29.2)	
5–10	53 (61.6)	108 (53.5)	
>10	6 (7.0)	28 (13.9)	
Oral sex			*p* = 0.217
Regular	7 (8.1)	28 (13.9)	
Occasionally	64 (74.4)	130 (64.4)	
Never	15 (17.4)	44 (21.8)	
Anal sex			*p* = 0.179 *
Regular	2 (2.3)	1 (0.5)	
Occasionally	12 (14.0)	40 (19.8)	
Never	72 (83.7)	161 (79.7)	
Age at onset of oral contraceptive			*p* = 0.544 *
Never used	7 (8.1)	17 (8.5)	
≤13 years	0 (0.0)	3 (1.5)	
14–16 years	36 (41.9)	81 (40.3)	
17–19 years	31 (36.0)	83 (41.3)	
≥20 years	12 (14.0)	17 (8.5)	
Contraception methods used previously			*p* = 0.032
Condom	**33 (36.7)**	**65 (27.2)**	
Oral contraceptive	**7 (7.8)**	**6 (2.5)**	
Intrauterine device	**17 (18.9)**	**44 (18.4)**	
None	**33 (36.7)**	**124 (51.9)**	
Smoking habits			*p* = 0.661 *
Not smoker	47 (54.7)	96 (47.8)	
1–10 cigarettes per day	23 (26.7)	61 (30.3)	
11–20 cigarettes per day	14 (16.3)	41 (20.4)	
>20 cigarettes per day	2 (2.3)	3 (1.5)	
Pack years of smoking			*p* = 0.685
Lower tertile (<2.5)	14 (38.9)	31 (33.0)	
Median tertile (<6.0)	10 (27.8)	34 (36.2)	
Upper tertile (>6.0)	12 (33.3)	29 (30.9)	
Alcohol use			*p* = 0.154
Yes	73 (85.9)	185 (91.6)	
No	12 (14.1)	17 (8.4)	
History of STDs			*p* = 0.038
STD history	**24 (26.7)**	**39 (16.3)**	
No STDs	**66 (73.3)**	**200 (83.7)**	
History of genital warts			*p* = 0.521
Yes	22 (25.6%)	58 (29.3)	
No	64 (74.4)	140 (70.7)	
Age at diagnosis of genital warts			*p* = 0.903 *
Never	64 (74.4)	142 (71.7)	
<20 years	9 (10.5)	27 (13.6)	
20–24 years	10 (11.6)	21 (10.6)	
>25 years	3 (3.5)	8 (4.0)	
Treatment of genital warts			*p* = 0.840 *
No treatment	12 (40.0)	29 (37.2)	
Topical treatment	6 (20.0)	25 (32.1)	
Electrocautery	1 (3.3)	3 (3.8)	
Cryotherapy	1 (3.3)	2 (2.6)	
Laser therapy	4 (13.3)	8 (10.3)	
Surgery	0 (0.0)	1 (1.3)	
Several treatments	6 (20.0)	10 (12.8)	

* Fisher’s exact test.

## Data Availability

Data and materials of this study are available from the corresponding author upon request.

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
