# Peer review of "Effect of a Second Pregnancy on the HPV Serology in Mothers Followed Up in the Finnish Family HPV Study"

_viruses, 2023, doi:10.3390/v15102109_

Round 1
Reviewer 1 Report
Suominen et al. have conducted a study titled "Effect of second pregnancy on HPV serology in mothers followed-up in the Finnish Family HPV Study." This research aimed to investigate the influence of pregnancy on human papillomavirus (HPV) natural antibody levels. They examined the seroprevalence and antibody levels of HPV 6, 11, 16, 18, and 45 in 89 women with a second pregnancy and 237 nonpregnant women over a 36-month follow-up period. Importantly, all participants were unvaccinated for HPV and pregnant at the study's outset.
The study collected serum samples from mothers at various time points, including baseline and during 12-month, 24-month, and 36-month follow-up visits. The key findings revealed that there were no statistically significant differences in mean antibody levels between women with second pregnancies and their nonpregnant counterparts. However, there were notable differences in serostatus, especially when the second pregnancy was ongoing at the 24-month mark.
Specifically, women with a second pregnancy were more likely to be seronegative for HPV 6, 11, 18, and 45 compared to nonpregnant women. Conversely, the opposite trend was observed for HPV16, with higher seropositivity rates among women with ongoing second pregnancies at the 36-month follow-up.
In summary, Suominen et al.'s study suggests that while second pregnancies may not significantly affect the levels of HPV antibodies, they may indeed influence serological outcomes in relation to specific HPV types. These findings contribute to our understanding of HPV serology in pregnant women, shedding light on the complex dynamics between pregnancy and HPV immunity.
The claims are properly placed in the context of the previous literature. The experimental data support the claims. The manuscript is written clearly enough that most of it is understandable to non-specialists. The authors have provided adequate proof for their claims, without overselling them. The authors have treated the previous literature fairly. The paper offers enough details of methodology so that the experiments could be reproduced.
Comments
1. In Table 2, the category "Number of deliveries" begins with the number 0 (zero). Since the study follows women with at least one pregnancy, it is reasonable to assume that no woman in the study has zero children.
2. Based on Table 2, I have calculated that the average number of deliveries for women with a second pregnancy in the follow-up is 1.13, while for women with no second pregnancy in the follow-up, it is 1.36. This discrepancy is expected, as women with four children have a lower probability of experiencing an additional pregnancy compared to women with only one child.
I am curious if there is a potential age difference between these two groups. Unfortunately, I could not locate the average age information within the table. Could you kindly provide the average age of women in each group?
3. In Table 2, there is a subtitle labeled 'Contraception method' with two options: 'Condom use' and 'Oral contraception,' and the total percentage adds up to 100%. Firstly, the proportion of condom users appears higher than expected. Secondly, I assume that some women in the study may not be using any contraception at all. Could you please clarify or provide an explanation?
Additionally, there should be 89 women with a second pregnancy, but the numbers only account for 57 women. Likewise, there are 237 nonpregnant women, but the numbers only represent 115 women. Could you please provide clarification or an explanation for this discrepancy?
4. In Table 2, there is a second subtitle labeled 'Intrauterine device.' I suggest classifying this as a contraception method, similar to 'Condom' and 'Oral contraception.' It would be beneficial to merge these subheadings into one category and ensure that the total percentages sum up to 100%, including women who do not use any contraceptives at all.
Reviewer 2 Report
A very interesting cohort study evaluating (among others) the possible effect of second pregnancy on HPV serology. The cohort monitoring started 25 years before and in this manuscript the authors base their results on 327 women, 89 in the group that had second pregnancy during follow up and the remaining 238 that did not.
A few comments that may improve this manuscript:
1. Line 67, states 329 women however this study focuses on 327, please clarify if these women were lost to follow up, as the reader becomes confused since 329 does not sum to the two groups populations (89+238).
2. In the statistical analysis, for completeness reasons, please clarify the test used for normality test, probably Kolmogorov Smirnoff?
3. In the results section suddenly appear three groups: a) women having an ongoing second pregnancy at the time of follow-up visit, b) women having a second pregnancy at some other timepoint, and c) not having a second pregnancy at all during the follow-up. Here are raised several questions: 1. What is the role of the second group? 2. Why is it different from the first group? 3. Did women of the second group had second pregnancy in the past? 4. are their data missing or not monitored? 5. Are the women of the second group included in the 89 ?, probably not, 6. Why you do not combine both groups since all women had 2nd pregnancies and this is the group of interest in the study? the role of the second group becomes unclear especially since it is not mentioned from the beginning, i.e. in the methods section. In my opinion this need clarification or to combine the two groups and mention it in the methods section (in my opinion this does not pose any study design issues).
4. Figure 1 does not allow easy comparison of the MFI among the HPV subtypes, could it fit in a single page? Perhaps putting two images in side-by-side arrangement during paper production?
5. In subsequent analysis (line 121 and below) it is not clear if studied women involve the women with 2nd pregnancy and follow up or all women with 2nd pregnancy (again referring to the three groups of figure 1)
6. Finally, there are some linguistic issues that result in comprehension issues, for example line 131: “When the second pregnancy was ongoing at 36 months, ….. “, seems that the second pregnancy had a 36 months duration!, you probably mean 36 months after the base line ? perhaps the authors mean the “second pregnancy group”
7. Otherwise, the data support the main conclusion that other factors apart from second pregnancy should be investigated as responsible for seropositivity, perhaps new infections?
Perhaps the authors may revise the manuscript for some phrases misleading the readers.
Reviewer 3 Report
GENERAL COMMENT:
The manuscript addresses a topic still not clearly understood such as the serological aspects of HPV infection. In particular, the authors focused on evaluating the impact of pregnancy on HPV antibody levels and serological status using a longitudinal approach, which represents the main strength of this study. Overall, the opinion on this article is particularly positive, as it is well-written and interesting to read. However, it could be improved by clarifying some points of the methodological approach, such as the characteristics of the patients’ cohort.
INTRODUCTION:
This section is well-written and clearly describes the objectives of the study, however, it appears to be slightly short. It could be extended by reporting more pieces of information about, for example, Human papillomaviruses, HPV infection, HPV antibodies, HPV vaccines etc.
I invite the authors to consider the following work:
G. Capra et al 2022, Human Papillomavirus (HPV) Infection and Its Impact on Male Infertility, Life (Basel).
G. Capra et al. 2017, Potential impact of a nonavalent vaccine on HPV related low-and high-grade cervical intraepithelial lesions: A referral hospital-based study in Sicily. Hum Vaccin Immunother
L. Bosco et al. 2021, Potential impact of a nonavalent anti-HPV vaccine in Italian men with and without clinical manifestations. Sci Rep
Page 2, lines 55-58: “Having enrolled in the cohort…”
This information is of utmost importance; however, I believe it should be placed in the “Participants” paragraph of “Materials and Methods”. In this section, you could simply refer to “unvaccinated women”.
MATERIALS AND METHODS:
I believe that the characteristics of the cohort could be better described. For example, although it could be deduced that all the participants were HPV-positive at some point in their lives, it is a point that should be clarified. In general, it should not be assumed that the readers know the characteristics of the Finnish Family HPV Study cohort.
Moreover, the groups in which the cohort is divided are described quite confusingly. In fact, if in the “Participants” paragraph two groups are depicted (second pregnancy during the follow-up vs. no second pregnancy during the follow-up), later, in the “Results” section, you talk about three groups (2nd pregnancy vs. 2nd pregnancy during follow-up vs. No follow-up pregnancy). It would be appropriate to report the same division into groups among the different paragraphs.
DISCUSSION:
Page 9, lines 178-186: “In this study, we observed that women who developed second pregnancy …”
You may put the description regarding demographic and clinical variables after the explanation of the antibody level results, so as to follow the order used in the "Results" section.
Page 9, lines 187-197: “In the present series, the mean antibody levels to HPV6, HPV11, HPV18 and HPV45…”
This period is clearly written; However, it comments on data not described in the "Results" paragraph, except in the graphs. I therefore consider it appropriate to report this information also in the text of the "Results" section.
Round 2
Reviewer 2 Report
The authors have addressed all review comments
In my opinion the manuscript can be published in current form.